# Effects of Different Levels of Yucca Supplementation on Growth Rates, Metabolic Profiles, Fecal Odor Emissions, and Carcass Traits of Growing Lambs

**DOI:** 10.3390/ani13040755

**Published:** 2023-02-19

**Authors:** Ibrahim S. Alsubait, Ibrahim A. Alhidary, Ahmed A. Al-Haidary

**Affiliations:** Department of Animal Production, College of Food and Agriculture Science, King Saud University, P.O. Box 2460, Riyadh 11451, Saudi Arabia

**Keywords:** *Yucca schidgera* extract, *Awassi lambs*, complete pelleted diet, performance, carcass traits

## Abstract

**Simple Summary:**

A complete pelleted diet is one of the most popular and effective approaches for feeding livestock because of its contribution to improved flock productivity, low cost, and profitability. However, there are a number of negative effects associated with feeding such diets to livestock which can impact the livestock producer and consumer preferences; these include wool biting, wool eating, odor emission, and the dark color of ruminal tissue. The addition of yucca plays a beneficial role in the nutrition and welfare of livestock, particularly sheep, by reducing the negative effects of feeding complete pelleted diets. Animal diets supplemented with yucca target growth, productivity, and physiological responses. The aim of this study was to improve the quality of complete pelleted diets in order to avoid or mitigate their effects on fecal and urinary odor emission. The results found that supplementing *Yucca schidgera* extract at a level of 600 mg YS/kg dry matter (DM) in feed improved the fecal and urinary odors of lambs fed on pelleted diets.

**Abstract:**

Sixty male *Awassi lambs* were used to investigate the effects of dietary *Yucca schidgera* extract (YS) on the production, fecal and urinary odor emissions, and carcass traits of growing lambs fed complete pellets. Lambs were fed either a complete pelleted diet without yucca (control) or supplemented with 300 or 600 mg YS/kg dry matter (DM) during the 84-day experiment. The weights and feed consumption of the lambs were measured weekly. Blood samples were taken on days 1, 28, 58, and 84, and ruminal fluid samples were collected on day 70. On day 90, the odor emissions from feces and urine were measured. On day 84, 12 lambs were slaughtered for the evaluation of carcass and meat quality. The final values for bodyweight, bodyweight gain, and feed efficiency of lambs fed the YS_300_ diet were 3.40%, 6.64%, and 6.17%, respectively, higher (*p* < 0.05) than those fed the YS_600_ diet. Additionally, the percentage of dressing, myofibril fragmentation index, and ruminal isovalerate percentage of lambs treated with YS_600_ were higher than those treated with YS_300_. Compared with the control, the addition of yucca reduced odor emissions from feces and urine. In conclusion, dietary YS_300_ had no additional benefits on growth rate, feed efficiency, and carcass traits, while dietary YS_600_ improved fecal and urinary odors.

## 1. Introduction

Extensive grazing systems do not always meet the nutrient requirements of the majority of livestock raised on them. Previous studies have successfully incorporated a variety of low-quality raw feed materials, such as cotton seed hulls, taramira oil cake, and sunflower hulls, into complete pelleted diets to fill this nutritional gap, resulting in economic gains for animal producers through improved animal growth rates, ruminal fermentation, and feed efficiency [1,2,3,4,5,6,7].

The use of natural feed additives in animal rations and the modification of feed formulations to improve the physical and microbiological processes of ruminal fermentation, thereby increasing the supply of volatile fatty acids (VFAs) and essential microbial proteins for ruminant growth [8,9], are gaining in popularity. These feed additives also play crucial roles in the regulation of toxic fermentation byproducts in the rumen [10,11]. There is a negative relationship between methane production and energy utilization in ruminants; therefore, the use of feed additives to reduce methane production and increase energy utilization is an attractive method of manipulating rumen fermentation [12,13].

Yucca is a medicinal plant that was grown originally in Mexico and the southwestern United States. Approximately 10% of the plant mass is dry matter [14,15]. Generally recognized as safe (GRAS), the US Food and Drug Administration has allowed the use of numerous YS products, including yucca extract and yucca powder, as additives in the food and drink industry, as well as in livestock and poultry feeds [16,17,18]. The biologically active surfactants in YS have been identified as steroid saponins and glycocomponents [18,19]. YS contains a number of polyphenolics which have antioxidant, anti-inflammatory, urease-inhibiting, and antiprotozoal characteristics, as well as properties related to the regulation of lipid metabolism [18,20,21,22]. Commercially, YS, as a feed additive, is used primarily in controlling environmental ammonia in livestock and poultry facilities in order to reduce odor emissions [23,24]. Several studies have indicated that the addition of YS powder to livestock feeds improves productivity and feed efficiency and increases ruminal VFA concentrations, as well as contributing to environmental control, microbial-activity modification, and blood biochemical parameters [23,24,25,26,27,28,29,30]. 

The effects of including dietary YS extract in completed pelleted diets on the productive performance of ruminant animals, particularly growing lambs, are largely unknown, and previous livestock nutrition studies have yielded contradictory results. Therefore, this study sought to determine whether feeding growing lambs on pelleted diets supplemented with varying levels of YS has beneficial effects on productivity, blood metabolic variables, ruminal fermentation, fecal and urine gas emissions, and carcass traits.

## 2. Materials and Methods

### 2.1. Experimental Design and Diets

The experiment was undertaken at the Animal Experimental Station, Department of Animal Production, College of Food and Agricultural Sciences, King Saud University, Riyadh, Saudi Arabia, and all procedures in this experiment followed the Animal Welfare Act of Practice for the Care and Use of Animals for Scientific Purposes, with approval from the King Saud University Research Ethics Committee (REC-KSU; Ethics Reference No.: KSU-SE-21-01). Sixty male Awassia lambs initially weighing 26.5, SD ± 2.0 kg, and aged between 3 and 4 months were purchased from the Riyadh livestock market and then transported to the Experimental Station. On the day of arrival, the animals’ weights were recorded and they were ear-tagged and treated against internal and external parasites and vaccinated against the commonest diseases. Animals were given a 14-day adaptation period, and, during this period, they were housed in shaded pens (4.0 m long, 3.0 m wide; 5 lambs in each pen). On the first day of the experiment, the lambs were randomly assigned to one of three treatments (four replicates for each treatment): (1) a complete pelleted diet (without YS supplementation; the basal diet; CON); (2) the basal diet with 300 mg YS/kg DM (YS_300_); and (3) the basal diet with 600 mg YS/kg DM (YS_600_). The *Yucca schidgera* extract BIOPOWDER^®®^ was provided by Agroin (Ensenada, BC, Mexico). All experimental feeds were formulated according to NRC recommendations [31] for growing lambs (Table 1). 

### 2.2. Growth Preformance and Feed Intake

The lambs were weighed on day 1 and then every two weeks in order to determine growth performance (including bodyweight gain and average daily gain). Feed intake was recorded weekly for each pen by calculating the difference between feed offered and feed refused. The feed conversion ratio (FCR) for each lamb was determined by subtracting dry-matter intake consumed from the average daily gain, and then represented as kg of DMI to kg of BW.

### 2.3. Blood Sample Processing and Analysis

Blood samples from all lambs were taken in the morning before feeding on days 1, 28, 58, and 84. Serum was obtained after the centrifugation of blood samples at 2400× *g* for 15 min at 4 °C and was then frozen at −20 °C until analysis. Commercial biochemical regents from Randox Laboratories (Antrim, UK) and a microplate reader (Multiskan EX, Thermo Fisher Scientific, Inc., Waltham, MA, USA) were used to determine the concentrations of glucose, total protein, albumin, urea, and creatinine in serum, following the manufacturers’ instructions.

### 2.4. Rumen Fermentation Profiles

Before the morning feeding, samples of rumen fluid (50 mL) from 15 lambs in each treatment were collected using an oral stomach tube on day 70 for evaluation of the ruminal fermentation profiles. The samples were analyzed for pH values, strained through four layers of cheesecloth, transferred into plastic tubes, acidified with 2 mL of concentrated sulfuric acid, and then frozen at −20 °C for further analysis. The ruminal fluid samples were then thawed and analyzed for ammonia (using a TECO Diagnostics kit) and volatile fatty acid (VFA) proportions and totals, including acetic, propionic, butyric, isobutyric, valeric, and isovaleric acids, as described by [32], using 2-ethylbutyric acid as an internal standard and a gas chromatograph–mass spectrometer (7010C Triple Quadrupole GC/MS, Agilent Technologies, Palo Alto, CA, USA). The total volatile fatty acid concentrations were expressed as concentrations mol/l (mM), while individual VFA proportions were expressed as molar percentages of total molar VFAs (% mol).

### 2.5. Fecal and Urinary Odor Evaluation

On day 90, seven lambs from each treatment were removed into metabolism cages (7 days) for the digestibility trial. During the collection period (3 days), urine and feces from each animal were measured daily. Representative samples of the collected urine and feces were used for the evaluation of fecal and urinary odor emissions. Fecal (150 g) and urine (150 mL) samples were mixed in 3 L plastic boxes, and then the mixtures of feces and urine were kept at room temperature for 0, 12, and 24 h to ferment and produce odor gases. After each fermentation period, the total ammonia, hydrogen sulfide, and acetic acid contents in the mixtures were quantified using colorimetric gas detector tubes (model: Gv100s Gas Detection System, Gas Tec, Wangara, WS, Australia) and a gas chromatograph–mass spectrometer (7010C Triple Quadrupole GC/MS, Agilent Technologies, Palo Alto, CA, USA), following a methodology described by [33].

### 2.6. Carcass Traits and Meat Quality

At the end of the experiment (day 84), 12 lambs from each treatment were slaughtered after 16 h of feed deprivation. With the live bodyweights obtained, hot carcasses and different organs were weighed immediately after slaughtering and then stored at 4 for 24 h. The carcasses were weighed again the next day for evaluation of the cold carcasses, and the percentages of dressing and chilling losses for each carcass were recorded. The right side of each carcass was separated to measure the pH value, back-fat thickness, the area of the longissimus thoracis muscle, and meat color value. Samples of muscles (about 300 g) were taken for the determination of water-holding capacity (WHC) [34], cooking loss (CL) [35], and myofibril fragmentation index (MFI) [36], while meat hardness, cohesiveness, chewiness, and springiness were measured using a texture analyzer (TA; HD, Stable Micro Systems, Surrey, UK) with a compression-plate attachment. 

### 2.7. Statistical Analyses

All data were analyzed using a complete randomized design with general linear model procedures of statistical analysis software (SAS Institute Inc., Cary, NC, USA). The statistical model included the level of yucca, the collection day, and the interaction of treatment and day as fixed effects, with animal within treatment (with the exception of using pens for feed intake data) as a random effect. Carcass and meat quality characteristics, ruminal fluid fermentation, and fecal–urinary odor emissions data were entered into the GLM model in SAS. Data are reported with the least-square means (±SEs), and differences between them were considered significant at *p* ≤ 0.05.

## 3. Results

### 3.1. Dry-Matter Intake (DMI), Average Daily Gain (ADG), and Feed Efficiency

The effects of the dietary treatments on the productive performance of the growing lambs are presented in Table 2. There were no differences (*p* > 0.05) in dry-matter intake (DMI) across the treatment groups. The mean DMI was similar for all the treatments (1.48 kg/day). In contrast, the final bodyweights, bodyweight changes, average daily gains, and feed efficiencies were lowest (*p* < 0.05) in the lambs in the YS_300_ group compared to the other groups.

### 3.2. Serum Biochemical Variables

Differences in the serum concentrations of the biochemical variables of growing lambs according to treatment are presented in Table 3. 

The serum concentrations of total protein, albumin, and urea did not differ (*p* > 0.05) between the dietary treatments. When compared to lambs fed the CON diet, the YS_300_ and YS_600_ diets resulted in a decrease (*p* = 0.01) in serum glucose concentration and an increase (*p* = 0.01) in serum urea concentration. When compared with the YS_300_ and CON groups, the YS_600_ group had intermediate glucose concentrations.

### 3.3. Rumen Fermentation Profile

The means of pH values and the concentrations of total VFAs and ammonia in the ruminal fluid of growing lambs fed with different levels of yucca are presented in Table 4. The overall pH value across the treatments was 6.05, ranging from 5.98 to 6.12. Total VFA concentrations and proportions of the majority of individual VFAs in ruminal fluid were not affected (*p* > 0.05) by different levels of yucca supplementation. When compared with lambs fed the CON diet, yucca supplementation at 300 mg YS/kg DM increased the isobutyrate molar percentage in the ruminal fluid of growing lambs, whereas yucca supplementation at 600 mg YS/kg DM increased isovalerate molar percentage in comparison with CON. In addition, lambs in the YS_600_ treatment group had (*p* < 0.05) the lowest ruminal ammonia concentrations compared with the other treatments (Table 4). 

### 3.4. Fecal and Urinary Odor Emissions

The effects of different levels of yucca dietary supplementation on the concentrations of fecal and urinary odor gases emitted by growing lambs are presented in Figure 1. The concentrations of all gases measured (acetic acid, hydrogen sulfide (H_2_S), ammonia (NH_3_), and carbon dioxide (CO_2_)) were affected (*p* < 0.05) by the yucca supplementation. The concentrations of NH_3_ and CO_2_ in lambs fed on YS diets (YS_300_ or YS_600_) were lower (*p* < 0.05) than those in lambs fed the CON diet. The addition of yucca at a level of 600 mg YS/kg DM to the lambs’ diets reduced (*p* = 0.03) the H_2_S concentration in the mixture (6.5 mg YS/kg DM) and increased (*p* = 0.01) the concentration of acetic acid in fecal and urinary odor emissions (11.3 mg YS/kg DM) compared to the lambs fed the complete pelleted diet without YS supplementation (CON group).

### 3.5. Carcass Characteristics and Meat Quality

The differences between the dietary treatments in terms of carcass characteristics and meat quality are presented in Table 5 and Table 6, respectively. In comparison with lambs in the YS_600_ group, lambs in the YS_300_ group had (*p* < 0.05) increased slaughter weights and shoulder percentages, which were similar to the values for lambs in the CON group. The YS_600_ group had higher chilling losses and lower shoulder percentages than the CON group. The dressing percentage was higher for the CON and YS_600_ groups compared with the YS_300_ group (*p* < 0.05). In addition, chilling losses were higher for the YS_600_ group compared with the CON group (*p* < 0.05), whereas these were intermediate for the YS_300_ group. There was no significant difference in organ weights between the treatments. The rack ribs and color components were not affected by the treatments (*p* > 0.05). Texture-profile analysis showed some significant differences according to the treatment. The myofibril fragmentation index (MFI) was significantly higher for the CON and YS_600_ groups compared with the YS_300_ group (*p* < 0.05). Shear force increased in the YS_300_ treatment group, but the difference was insignificant when compared with the YS_600_ group (*p* < 0.05). Generally, the meat samples from the YS_300_ and YS_600_ groups were harder, springier, and chewier (*p* < 0.05). Other parameters, such as cooking loss, WHC, and cohesiveness, were not affected by the treatments (*p* > 0.05).

## 4. Discussion

In Saudi Arabia, sheep production is a primary livestock industry in the agricultural sector. It plays a critical role in social and economic sustainability. According to [37], the sheep population in Saudi Arabia was estimated to be approximately 17.5 million head, representing approximately 70% of the total livestock population [37]. Under extensive grazing systems, sheep often do not receive their required nutrients because of shortages in natural resources, and this can cause reductions in the productive efficiency of flocks. Providing a range of feedstuffs, in particular forage, required by feeding ruminants is currently a major challenge facing the livestock industry in Saudi Arabia because of high feed prices.

According to [38], Awassi lambs fed complete pelleted diets had 10 and 21% higher ADGs and feed efficiencies, respectively, when compared to those fed on grain barley and alfalfa hay. Previous research found that feeding lambs a complete pelleted diet resulted in an increase in the color of ruminal tissue (L* value from 50.66 to 29.11) and a reduction in the pH values of ruminal fluid from 6.05 to 5.27 [39,40].

Saponins, a group of secondary compounds in plants, are able to alter rumen fermentation characteristics and encourage animal productivity [41]. Several studies have indicated that yucca saponins increased productive indicators, such as growth rate, meat, and milk production, in sheep when they were fed roughage diets [42,43]. A study reported in [25] observed that growing lambs fed a diet supplemented with YS at levels between 40 and 60 mg YS/kg DM had higher growth rates [25]. This is in agreement with the results obtained in our study: YS improved bodyweight, ADG, and feed efficiency ratio, in particular at 0 and 300 mg YS/kg DM. YS_0_ and YS_300_ generally performed better than YS_600_. Overall, lambs treated with YS_300_ had a higher final bodyweight (3.4%), ADG (6.6%), and feed efficiency (6.17%) in comparison with YS_600_ lambs.

Serum concentrations of total protein, albumin, and creatine were not affected by yucca supplementation in this study, and were all within the reference ranges reported in [44]. These results are consistent with those of previous studies [45,46,47] showing that neither the levels nor saponins of yucca supplementation affected total protein, albumin, and creatine levels in ruminants. A reduction in serum urea concentration was associated with the highest level of yucca supplementation (YS_600_) in our study, and the mean urea concentration was within the normal range for sheep (3.67–9.28 mM) [48]. Previous studies have indicated that yucca extract components might modify kidney function by increasing the rate of urea clearance and lowering blood urea concentrations [26].

In the current study, there was no response to the addition of yucca in terms of total VFA concentrations or in the proportions of the majority of individual VFAs in ruminal fluids. This is consistent with the results of previous studies, which indicated that dietary yucca supplementation had no beneficial effects on the VFA profiles of ruminant animals [49,50]. The authors of [51] suggested that saponins supplemented at higher concentrations might adversely affect rumen fermentation. Conversely, a report by [52] observed increases in gas and VFA production in dairy cattle fed a concentrate diet supplemented with yucca.

Reductions of 5.0 and 18.5% in ruminal ammonia concentrations were observed in lambs fed diets supplemented with 300 and 600 mg YS/kg DM, respectively, compared to those fed the CON diet (without yucca) in the current study. The complete pelleted diets of 300 and 600 mg YS/kg DM produced less ammonia compared with the control diet. The mechanism by which ammonia concentration is affected by the addition of YS could be attributed to an increase in ruminal nitrogen, as metabolic pathways (protein digestibility) and the involvement of saponins appeared to assist in reducing ammonia production in the rumen when yucca was added to the diets, while rumen ammonia concentration increases as a result of proteolysis in protein bacteria due to increases in rumen bacterial digestion by protozoa [50,53]. Thus, a reduction in ammonia production in the rumen could be attributed to the antiprotozoal activity utilized by saponins [53,54]. Similar findings have been reported in several previous studies [26,27,55].

Yucca has important applications in ruminant nutrition and plays an important role in the regulation of fecal and urinary odor emissions, this being one of the main reasons for adding it to ruminant diets. This role is related to its involvement in various physiological functions, including the improvement of ruminal fermentation characteristics and microbiome responses and alterations to digestion and metabolic pathways, and the high-value compounds present in yucca (e.g., steroid saponins, glycocomponents, phenolics, and several enzymes). For example, yucca plays beneficial roles in enhancing nitrogen utilization through digestion, absorption, and metabolic and excretion processes in ruminants and poultry [56,57,58]. These roles could be attributed to the reductions in the concentration of ammonia in fecal and urinary odor emissions that were observed in the current study when lambs were fed complete pelleted diets supplemented with yucca at a level of 600 mg/kg. Meanwhile, the release of fecal and urinary odors with the YS addition in the current study would have been due to the increase in acetic acid, which is involved in the regulation of a variety of enzymes; these include mono-oxygenases and the enzymatic family of aminotransferases. These enzymes are mainly responsible for the conversion of tryptophan in the diet into acetic acid [58].

Differences in carcass characteristics and meat quality between the dietary treatments showed that feeding lambs the 300 mg YS/kg DM yucca diet increased slaughter weight, hot-carcass yield percentage, and shoulder wholesale cut weight. The dressing percentage increased with the 600 mg YS/kg DM yucca diet, and chilling loss was reduced with the CON diet. In general, the meat samples from the yucca-supplemented groups were harder, springier, and chewier. The results obtained here are in disagreement with those obtained by the authors of [59], the latter reporting insignificant differences in the weights of parts of the carcass, meat, fat, and bones of lambs fed YS extracts at incremental levels from 0 to 400 mg YS/kg DM. In broilers fed with YS (100 mg YS/kg DM) supplements, the authors of [60] reported positive improvements in live weight and carcass characteristics, such as carcass yield, dressing percentage, and breast. Similarly, these results were confirmed by previous research, which found that YS supplementation of broilers improved evisceration weight and yield of breast and thigh meat [61,62]. The mechanism by which the carcass and meat quality characteristics are affected positively by the addition of YS into livestock diets could be attributed to the improvements in gut histomorphology and nutrient absorption and utilization caused by steroid saponins [60].

## 5. Conclusions

Under the conditions of the current study, the results indicated that the inclusion of dietary YS at a level of 300 mg YS/kg DM in the diet of growing lambs had no additional consistent effect on growth rate, feed efficiency, carcass yield, or meat quality. Although, there were reductions in growth rate and carcass traits associated with YS supplementation with 600 mg YS/kg DM, the high level of YS supplementation resulted in reductions in ammonia concentrations in ruminal fluids and fecal odor emissions. More research is required to clearly define the effects of YS supplementation and the mechanisms responsible for these effects on productivity and gas emissions. These studies should be considered to determine the most efficient sources and levels of YS supplementation for feeding sheep. 

## Figures and Tables

**Figure 1 animals-13-00755-f001:**
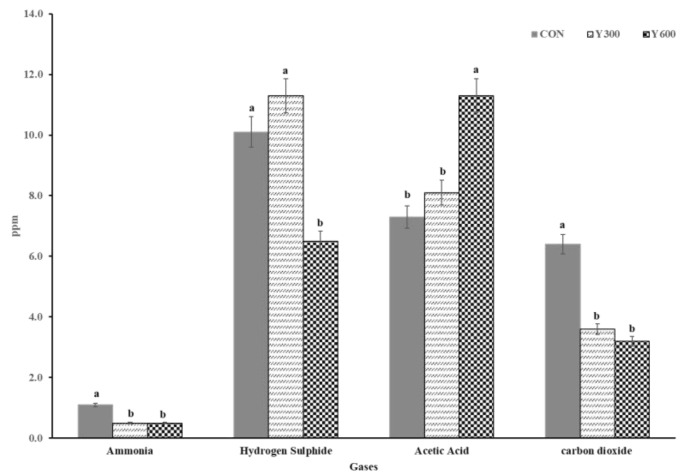
Effect of dietary supplementation with different levels of yucca on fecal and urinary odor emissions of growing lambs. Values are means of growing lambs (*n* = 21) for 84 days. CON = a complete pelleted diet (the basal diet without YS supplementation); YS_300_ = the complete pelleted diet supplemented with YS at a level of 300 mg YS/kg DM; and YS_600_ = the complete pelleted diet supplemented with YS at a level of 600 mg YS/kg DM, Means without a common letter (a, b) differ (*p* < 0.05).

**Table 1 animals-13-00755-t001:** Ingredients and chemical composition of the basal diet used in the experiment ^1^.

Item	Content
Ingredients, % of dietary dry matter	
Corn, grain	29.92
Feed wheat, grain	18.40
Alfalfa hay	9.10
Palm kernel meal	11.40
Soybean hulls	12.03
Wheat bran	12.10
Salt	0.47
Limestone	2.58
Molasses	3.85
Mineral and vitamin premix ^2^	0.15
Nutrient composition, dry-matter basis	
Dry matter, (%)	92.43
Ash, %	7.79
Crude protein, %	14.51
Ether extract, %	3.98
Neutral detergent fiber, %	33.23
Acid detergent fiber, %	20.12
Metabolizable energy, MJ/kg	11.6

^1^ The basal diet was formulated and provided as a complete pelleted diet (ARASCO Manufactory, Riyadh, Saudi Arabia). ^2^ Containing, per kg: 10,000 IU vitamin A, 1000 IU vitamin D, 20 IU vitamin E, 300 mg, Mg, 24 mg Cu, 0.6 mg Co, 1.2 mg I, 60 mg Mn, 0.3 mg Se, and 60 mg Zn.

**Table 2 animals-13-00755-t002:** Mean bodyweight (BW), bodyweight gain (BWG), average daily gain (ADG), dry-matter intake (DMI), and feed efficiency (G:F ratio) of growing lambs fed complete pelleted diets supplemented with different levels of yucca.

Item	Dietary Treatments ^1^			SE	*p*-Value
	CON	YS_300_	YS_600_		
Initial BW, kg	26.26	26.83	26.97	1.462	0.56
Final BW, kg	45.78 ^a,b^	46.25 ^a^	44.68 ^b^	1.828	0.04
BW change, kg	18.56 ^a^	18.58 ^a^	17.30 ^b^	0.674	0.03
ADG, g/d	240 ^a^	241 ^a^	225 ^b^	18.213	0.03
DMI, kg/d	1.48	1.49	1.48	0.276	0.43
G:F ratio	162 ^a^	162 ^a^	152 ^b^	7.208	0.02

^a,b^ Within a row, means without a common superscript differ (*p* < 0.05). ^1^ Values are for growing lambs (*n* = 60). CON = a complete pelleted diet (the basal diet without YS supplementation); YS_300_ = the complete pelleted diet supplemented with YS at a level of 300 mg YS/kg DM; and YS_600_ = the complete pelleted diet supplemented with YS at a level of 600 mg YS/kg DM. SE: standard error.

**Table 3 animals-13-00755-t003:** Effects of different levels of yucca supplementation on serum concentrations of biochemical variables of growing lambs.

Item	Dietary Treatments ^1^			SE	*p*-Value
	CON	YS_300_	YS_600_		
Glucose, mM	2.11 ^a^	1.78 ^b^	1.72 ^b^	0.283	0.02
Total protein, g/L	53.17	60.66	58.89	6.667	0.23
Albumin, g/L	29.00	28.79	29.30	1.522	0.34
Urea, mM	5.99 ^c^	7.11 ^a^	6.60 ^b^	0.459	0.01
Creatinine, µM	104.3	106.7	105.3	2.207	0.09

^a–c^ Within a row, means without a common superscript differ (*p* < 0.05). ^1^ Values are for growing lambs (*n* = 60). CON = a complete pelleted diet (the basal diet without YS supplementation); YS_300_ = the complete pelleted diet supplemented with YS at a level of 300 mg YS/kg DM; and YS_600_ = the complete pelleted diet supplemented with YS at a level of 600 mg YS/kg DM. SE: standard error.

**Table 4 animals-13-00755-t004:** Effects of different levels of yucca supplementation on the ruminal fermentation profiles of growing lambs.

Item	Dietary Treatments ^1^			SE	*p*-Value
	CON	YS_300_	YS_600_		
pH value	6.08	5.98	6.12	0.342	0.67
Total VFAs, mM	53.95	48.9	49.13	4.568	0.16
Acetate, % mol	32.80	32.93	32.84	2.343	0.56
Propionate, % mol	41.74	41.03	39.72	2.519	0.25
Butyrate, % mol	18.49	18.57	19.30	1.457	0.12
Isobutyrate, % mol	1.37 ^b^	2.16 ^a^	1.29 ^b^	0.872	0.05
Valerate, % mol	2.71	2.93	3.68	0.461	0.53
Isovalerate, % mol	2.89 ^a,b^	2.38 ^b^	3.16 ^a^	0.563	0.04
Acetate: propionate ratio	0.79	0.80	0.83	0.121	0.44
Ammonia, µM	6.32 ^a^	6.02 ^b^	5.33 ^c^	0.276	0.01

^a–c^ Within a row, means without a common superscript differ (*p* < 0.05). ^1^ Values are for growing lambs (*n* = 45). CON = a complete pelleted diet (the basal diet without YS supplementation YS_300_ = the complete pelleted diet supplemented with YS at a level of 300 mg YS/kg DM; and YS_600_ = the complete pelleted diet supplemented with YS at a level of 600 mg YS/kg DM. SE: standard error.

**Table 5 animals-13-00755-t005:** Effects of different levels of dietary yucca supplementation on the carcass characteristics of growing lambs.

Item	Dietary Treatments ^1^			SE	*p*-Value
	CON	YS_300_	YS_600_		
Carcass profile					
Slaughter BW, kg	45.10 ^a^	46.70 ^a^	44.60 ^b^	1.123	0.04
Hot carcass, kg	21.93	21.79	21.41	1.162	0.16
Cold carcass, kg	21.51	21.28	20.80	1.148	0.34
Dressing, %	48.68 ^a^	46.65 ^b^	48.06 ^a^	0.819	0.03
Chilling losses, %	1.95 ^b^	2.33 ^ab^	2.85 ^a^	0.362	0.02
Organ weight, kg					
Non-carcass components ^2^	6.92	6.58	6.93	0.232	0.23
Liver	0.75	0.82	0.77	0.082	0.56
Heart	0.13	0.13	0.14	0.014	0.34
Kidneys	0.12	0.13	0.11	0.019	0.67
Stomach	1.44	1.20	1.64	0.412	0.11
Tail	2.74	2.56	2.67	0.557	0.09
Wholesale cut weight, kg					
Shoulder	2.60 ^a^	2.70 ^a^	2.49 ^b^	0.117	0.02
Rack	0.96	0.98	0.94	0.048	0.57
Loin	1.25	1.29	1.20	0.113	0.18
Leg	2.95	2.94	2.97	0.138	0.56
Foreshank and breast	1.20	1.16	1.15	0.059	0.22

^a,b^ Within a row, means without a common superscript differ (*p* < 0.05). ^1^ Values are for growing lambs (*n* = 36). CON = a complete pelleted diet (the basal diet without YS supplementation); YS_300_ = the complete pelleted diet supplemented with YS at a level of 300 mg YS/kg DM; and YS_600_ = the complete pelleted diet supplemented with YS at a level of 600 mg YS/kg DM. SE: standard error. ^2^ Non-carcass components = internal organs + visceral fat deposits + head + gastrointestinal tract.

**Table 6 animals-13-00755-t006:** Effects of different levels of dietary yucca supplementation on meat quality of growing lambs.

Item, Unit ^2^	Dietary Treatments ^1^			SE	*p*-Value
	CON	YS_300_	YS_600_		
Rack ribs					
Rib eye area, cm^2^	11.22	9.24	9.54	2.321	0.14
Body wall fat, mm	5.89	6.33	5.20	0.317	0.21
Back fat, mm	4.70	4.69	4.42	0.209	0.12
Color components					
L *	26.61	28.93	27.91	1.118	0.37
a *	15.53	14.95	14.28	1.433	0.56
b *	3.40	3.52	3.64	0.262	0.67
Visceral fat deposits, kg					
Total	1.15	0.97	0.85	0.271	0.09
Omental	0.53	0.51	0.45	0.054	0.67
Mesentery	0.31	0.15	0.13	0.106	0.23
Pericardial	0.03	0.05	0.03	0.138	0.56
KKCF	0.28	0.26	0.24	0.058	0.16
Texture profile analysis					
MFI	82.86 ^a^	70.06 ^b^	82.82 ^a^	2.557	0.02
Cooking loss, %	40.99	42.69	40.81	0.832	0.24
Shear force, kg	3.51 ^b^	4.53 ^a^	4.05 ^ab^	0.538	0.03
WHC	0.31	0.28	0.29	0.069	0.54
Hardness	0.43 ^b^	0.83 ^a^	0.76 ^a^	0.272	0.01
Springiness	0.51 ^b^	0.59 ^a^	0.55 ^ab^	0.048	0.02
Cohesiveness	0.48	0.48	0.50	0.051	0.19
Chewiness	1.07 ^b^	2.09 ^a^	2.27 ^a^	0.954	0.01

^a,b^ Within a row, means without a common superscript differ (*p* < 0.05). ^1^ Values are for growing lambs (*n* = 36). CON = a complete pelleted diet (the basal diet without YS supplementation); YS_300_ = the complete pelleted diet supplemented with YS at a level of 300 mg YS/kg DM; and YS_600_ = the complete pelleted diet supplemented with YS at a level of 600 mg YS/kg DM. SE. ^2^ KKCF = kidney knobs channel fat; MFI = myofibril fragmentation index; WHC = water-holding capacity. SE: standard error.

## Data Availability

The data and analyses presented in this paper are freely available from the corresponding author on reasonable request.

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
