# Peer review of "Effects of Different Levels of Yucca Supplementation on Growth Rates, Metabolic Profiles, Fecal Odor Emissions, and Carcass Traits of Growing Lambs"

_animals, 2023, doi:10.3390/ani13040755_

Round 1

Reviewer 1 Report

Authors should insert a hypothesis in their introduction.

The carcass and meat quality assessment methodology should be better detailed. The setting used for texture profiling analysis must be entered.

There is no discussion of the results obtained. The authors only conceptualize and report that other authors found similar and/or higher/lower values. It is necessary to make a connection between the results found and provide justifications for the answers obtained. The why should be explained.

It is not interesting that the authors compare their results with studies carried out with non-ruminants. Look for relevant articles and that ruminant animals were used in your study.

References must be in line with the journal's rules.

Author Response

We would like to thank you for considering our manuscript no. animals-2079410 entitled “Effects of different levels of yucca supplementation on growth performance, metabolic profile, fecal emission and meat quality of growing lambs fed pelleted diets”. We have revised the manuscript based on the required comments. Our responses to your comments are attached.

Reviewer 2 Report

The conclusion that inclusion of YS improved growth, feed efficiency, and carcass weight of growing lambs is questinalble as the differences between the control and YS300 were insignificant.

Author Response

(The authors gave the same response as above.)

Reviewer 3 Report

The authors conducted a trial on lambs fed Yucca to test growth performance, feed conversion indexes, carcass and meat yield and quality, and odour emission. Firstly, the manuscript is not editorially accurate and the English is of low quality. Furthermore, many statements are not in line with the results obtained. Finally, the statistical analysis does not seem to consider the experimental unit, which is the cage and not the individual animal. The work should be resubmitted after a thorough review by the authors. Punctual comments are given in the text of the manuscript in the form of post-its

Author Response

(The authors gave the same response as above.)

Reviewer 4 Report

The work is very interesting and carefully prepared. although the additive used is unlikely to be applicable in European conditions, the problem addressed is important. The research methods are appropriate and the approach is comprehensive.

Some issues arise:

whether supplementation with yucca extract 300 or 600 mg YS/kg dry matter (DM) was the authors' original idea or was it a result of previous experience?

whether the addition the YS powder is economically justified?

The impact part needs some explanation the effects of different levels of yucca dietary supplementation on the concentrations of fecal and urinary odor gases:

 The statement “Previous studies have demonstrated significant associations between the addition of yucca and the concentrations of gases emitted as fecal and urinary odors from ruminants and poultry” is too general.

“production reduced by 15% when dairy cattle were fed yucca–saponin” - production of what? which gases?

Please expand the topic of acetic acid because his growth after supplementation was significant.

good luck

Author Response

(The authors gave the same response as above.)

Round 2

Reviewer 3 Report

I have included the comments directly in the authors' response letter

Author Response

Dear Dr

We would like to thank you for reviewing our manuscript no. animals-2079410 entitled “Effects of different levels of yucca supplementation on growth performance, metabolic profile, fecal emission and meat quality of growing lambs fed pelleted diets”. We have revised the manuscript based on the required comments. Please have looked at our responses to your comments below:
